# The Influence of Cyclical Ketogenic Reduction Diet vs. Nutritionally Balanced Reduction Diet on Body Composition, Strength, and Endurance Performance in Healthy Young Males: A Randomized Controlled Trial

**DOI:** 10.3390/nu12092832

**Published:** 2020-09-16

**Authors:** Pavel Kysel, Denisa Haluzíková, Radka Petráková Doležalová, Ivana Laňková, Zdeňka Lacinová, Barbora Judita Kasperová, Jaroslava Trnovská, Viktorie Hrádková, Miloš Mráz, Zdeněk Vilikus, Martin Haluzík

**Affiliations:** 1Department of Sports Medicine, First Faculty of Medicine and General University Hospital, 12000 Prague, Czech Republic; kysel@palestra.cz (P.K.); dhalu@lf1.cuni.cz (D.H.); radka.petrakovadolezalova@vfn.cz (R.P.D.); 2Centre for Experimental Medicine, Institute for Clinical and Experimental Medicine, 12000 Prague, Czech Republic; laki@ikem.cz (I.L.); lacz@ikem.cz (Z.L.); troj@ikem.cz (J.T.); 3Diabetes Centre, Institute for Clinical and Experimental Medicine, 12000 Prague, Czech Republic; kapb@ikem.cz (B.J.K.); hrav@ikem.cz (V.H.); mrzm@ikem.cz (M.M.); 4Institute of Medical Biochemistry and Laboratory Diagnostics, First Faculty of Medicine, Charles University and General University Hospital, 12000 Prague, Czech Republic

**Keywords:** body composition, ketogenic diet, strength parameters, endurance, training

## Abstract

(1) Background: The influence of ketogenic diet on physical fitness remains controversial. We performed a randomized controlled trial to compare the effect of cyclical ketogenic reduction diet (CKD) vs. nutritionally balanced reduction diet (RD) on body composition, muscle strength, and endurance performance. (2) Methods: 25 healthy young males undergoing regular resistance training combined with aerobic training were randomized to CKD (*n* = 13) or RD (*n* = 12). Body composition, muscle strength and spiroergometric parameters were measured at baseline and after eight weeks of intervention. (3) Results: Both CKD and RD decreased body weight, body fat, and BMI. Lean body mass and body water decreased in CKD and did not significantly change in RD group. Muscle strength parameters were not affected in CKD while in RD group lat pull-down and leg press values increased. Similarly, endurance performance was not changed in CKD group while in RD group peak workload and peak oxygen uptake increased. (4) Conclusions: Our data show that in healthy young males undergoing resistance and aerobic training comparable weight reduction were achieved by CKD and RD. In RD group; improved muscle strength and endurance performance was noted relative to neutral effect of CKD that also slightly reduced lean body mass.

## 1. Introduction

The last decade has been characterized by the search for alternative dietary ways to achieve optimal body composition while maintaining or improving physical fitness and sports performance to promote healthy lifestyle and prevent chronic diseases [1,2]. Current trends in sports nutrition are increasingly reaching for the minimization of the carbohydrate component with ketogenic diet becoming a very popular approach, in particular in endurance athletes [3,4].

According to current definitions, carbohydrate intake within the range of 50–150 g per day can be described as non-ketogenic low-carbohydrate regimens [5]. Ketogenic diet is most commonly defined by a daily carbohydrate intake below 50 g per day or energy provision from carbohydrates for up to 10% of total energy intake [6]. Out of the frequently used approaches, targeted ketogenic diet allows carbohydrates to be consumed immediately around exercise to sustain performance without affecting ketosis [7]. The cyclical ketogenic diet (CKD) alternates periods of ketogenic dieting with periods of high-carbohydrate consumption [8]. The period of high-carbohydrate eating is supposed to refill muscle glycogen to sustain exercise performance [9].

The influence of ketogenic diets on sports performance is still the topic of an ongoing debate [10,11] with often conflicting results [12]. The overreaching mainstream nutrition philosophy for endurance athletes emphasizes a carbohydrate-dominant, low fat paradigm. Under these dietary conditions, carbohydrates are utilized as predominant fuel source to cover high volumes of aerobic exercise [13]. The appeal of low carbohydrate high fat diet for endurance athletes is likely due to the shift in fuel utilization, from a carbohydrate-centric model with limited glycogen sources to predominant fat utilization with much bigger and longer-lasting fat stores [14]. This metabolic shift, seen after a period of dietary alteration, is often referred to as being “fat-adapted”, which has been well-documented in studies since the 1980s [15]. Substantial reduction in carbohydrate intake promotes utilization of ketones and, according to some studies, it may enhance physical performance due to minimizing the reliance of body metabolism on carbohydrates [16,17] and reduce lactate deposition leading to enhanced recovery [18]. Importantly, ketogenic diets are, in particular in the short-term run, a very efficacious way to reduce body weight not only in physically active subjects but also in patients with obesity, type 2 diabetes and other chronic lifestyle diseases [19,20,21]. Nevertheless, it has to be noted that long-term compliance and efficacy of ketogenic diet is not optimal and most of the studies had rather limited duration [19,22].

Here we performed a randomized controlled trial to compare the effect of the cyclical ketogenic reduction diet (CKD) vs. nutritionally balanced reduction diet (RD) on body composition, muscle strength, and endurance performance in healthy young males undergoing regular resistance training three times/week combined with aerobic training three times/week. We hypothesized that CKD will be more efficacious in inducing fat loss as compared to RD while maintaining aerobic performance. To this end, we explored the effect of eight weeks of CKD vs. RD combined with regular exercise on body composition, and measures of strength and aerobic performance.

## 2. Materials and Methods

Twenty-five males of various fitness levels with minimum of one-year experience in resistance training recruited from colleges of physical education and through a website with readers interested in fitness and diets. Inclusion criteria were as follows: age between 18 and 30 years and a minimum one-year experience with resistance and aerobic training. Subject recruitment began in April 2019 and lasted until January 2020. Persons interested in participating were screened to ascertain they meet the minimum criteria for the enrollment into the study.

Exclusion criteria were current injuries or health conditions that might have affected sports performance or put them at risk for further injuries including the presence of cardiovascular diseases, diabetes mellitus, arterial hypertension, or any other diseases that required pharmacological treatment. Additionally, subjects taking any performance enhancing supplements (i.e., creatine, β-hydroxy β-methyl butyrate, caffeine, protein powder, weight gainer, thermogenics, etc.), were required to discontinue consumption at least one week prior to baseline testing and continue abstaining from their use for the remainder of the study. The study was approved (ethic approval code 764/18 S-IV)by the Human Ethics Review Board, First Faculty of Medicine and General University Hospital, Prague, Czech Republic and was performed in agreement with the principles of the Declaration of Helsinki as revised in 2008. Prior to randomization, all subjects were required to sign an informed consent.

Using electronic randomization system, subjects were randomly assigned to follow either a CKD or RD (both with total caloric intake reduction by 500 kcal/day) while participating in three strength workouts and three aerobic workouts per week (30 min run, heart rate around 130–140 beats/min.) for 8 weeks. Total caloric intake reduction by 500 kcal/day is counted from balanced hypocaloric diet with a reduction of energy intake by 500 to 1000 kcal from the usual caloric intake. The U.S. Food and Drug Administration (FDA) recommends such diets as the “standard treatment” for clinical trials (FDA, 1996)

Subject randomization and follow-up during the study is depicted in CONSORT diagram in Figure 1.

### 2.1. Baseline and Postinterventional Testing

Data collection during baseline and post-intervention testing included medical history, anthropometric examination, power performance test, bicycle spiroergometry, and blood drawings to obtain laboratory data. BMI was calculated by the scale, using the height measurement. Accurate height was measured using a basic stadiometer (Seca 222, Seca Co., London, UK).

#### 2.1.1. Biochemical and Anthropometric Examination

At baseline, all subjects were weighted, and their BMI was calculated. Body composition was measured using InBody Body Composition Analyzers (InBody230, InBody Co., Ltd., Seoul, Korea). Body weight and other body composition measurements (lean body mass, body fat mass, BMI, water content, and percentage of body fat) were taken with minimal clothes, no shoes, and measured to the nearest 0.5 kg.

Blood samples for biochemical measurements were taken prior to initiation of study and at the end of the study after 8 weeks of diet. Serum was obtained by centrifugation and samples were subsequently stored in aliquots at −80 °C until further analysis. The maximal storage time was 8 months.

Biochemical parameters liver test, urea, creatinine, and circulating lipids were measured to exclude liver, kidney, or lipid disorder. Creatine kinase and lactate dehydrogenase were measured to explore a possible influence of the diets on muscle regeneration. β-hydroxy-butyrate was measured to confirm a compliance to ketogenic diet.

β-hydroxy-butyrate was measured using TECOM Analytical Systems (TECOM Analytical Systems CS spol. s r.o., Prague, Czech Republic). Other biochemical parameters were measured by spectrophotometric methods using ARCHITECT c Systems device (Abbott Park, IL, USA.) in the Department of Biochemistry of the Institute for Clinical and Experimental Medicine in Prague.

#### 2.1.2. Strength and Aerobic Performance Testing

Power and performance testing were conducted over a 5-day period. Subjects signed for an hour block to participate in each test. Each block had a maximum of 5 subjects in a gym and the spiroergometry was reserved for each of them for an hour. Subjects were instructed to arrive at the gym 30 min prior to testing times and not to train for at least 24 h before testing. A strength performance testing for power output in the three exercises—bench-press, lat pull-down, and leg-press was performed as follows: The subjects underwent an adequate warm up. After resting for two to four minutes the subjects than performed a one-repetition maximum attempt of each exercise with proper technique. If the lift/press was successful, after resting for another two to four minutes the load was increased by 5–10% and another lift/press was attempted. If the subject failed to perform the lift/press, after resting for two to four minutes they attempted the lift/press with weight reduced by 2.5–5%.

#### 2.1.3. Methodology of Strength Testing

Upon arrival, the primary researcher explained the testing procedures and protocols and demonstrated each test. Subjects were instructed to warm up. Power and aerobic performance test administrators and personal researchers were blinded to the randomized group allocations. Each proband participated in bench press, lat pull-down, and leg-press to assess the maximum power performance.

#### 2.1.4. Aerobic Performance Testing

Aerobic performance testing was carried out by bicycle spiroergometry using analyzer of respiratory gases (Quark CPET, 1850 Bates Ave, Concord, CA 94520, Cosmed, USA). This metabolic cart measures expired airflow by means of a pneumotach connected to the mouthpiece. A sample line is connected to the pneumotach from which air is continuously pumped to O_2_ and CO_2_ gas analyzers. Prior to testing, the pneumotach was calibrated with six samples from a 3 L calibration syringe. The gas analyzers were also calibrated before each test to room air and calibration gases (15.21% O_2_ and 5.52% CO_2_, respectively). Heart rate (HR) was continuously recorded during exercise by electrocardiography (Fukuda Denshi FX-8322 Cardimax ECG, 17725 N. E. 65th Street Bldg. C, Redmond, WA. 98052 USA).

Prior to exercise, the subjects were instructed to maintain a pedal cadence between 70 and 90 rpm during exercise and to exercise to volitional fatigue. We used a modified exercise step protocol 0.33 W.min^−1^ as described by Gordon et al. [23]. The test was terminated when the subject was unable to maintain a pedaling cadence of 40 rpm.

Maximal oxygen consumption was assessed by the attainment of the following criteria: (1) a plateau (ΔVO_2_ ≤ 50 mL/min at VO_2_ peak and the closest neighboring data point) in VO_2_ with increases in external work, (2) maximal respiratory exchange ratio (RER) ≥ 1.10, and (3) maximal HR within 10 b/min of the age-predicted maximum (220-age). All subjects met the first two criteria.

Breath-by-breath gas exchange data from all tests were transferred to a spreadsheet program (MS Excel 365) for further analysis. In addition, data from the VO_2_max tests were time-averaged using 10 s intervals to examine the incidence of an oxygen plateau.

### 2.2. Diet Protocol

Subjects were randomly assigned by electronic randomization system to either CKD or RD group for 8 weeks. Subjects had a mandatory dietary session with a nutritionist prior to the beginning of the study which provided detailed instructions on accurately keeping dietary food intake records. All food record data were entered and analyzed using the DietSystem application (DietSystem App, DietSystem App, s.r.o., Czech Republic).

#### 2.2.1. Cyclical Ketogenic Reduction Diet

Total intake of energy was assigned to each participant based on lifestyle (individually calculated according to somatotype, physical activity, type of work, etc.) and was reduced by 500 kcal per day. Five days of low-carbohydrate phase, nutrient ratio (carbohydrates up to 30 g; proteins 1.6 g/kg; fats: calculation of energy intake instead of carbohydrates) in order to induce and maintain ketosis. Following with 2 days of carbohydrate phase (weekends): nutrient ratio (carbohydrates 8–10 g/1 kg of non-fat tissue, 70% intake; proteins 15%; and fat 15%).

#### 2.2.2. Reduction Diet

Principles of healthy nutrition, nutrient ratio (carbohydrates 55%, fat 30%, proteins 15% of total energy intake). The overall caloric intake (individually calculated according to somatotype, physical activity, type of work, etc.) was reduced by 500 kcal per day.

Both groups were given detailed instructions on acceptable foods for both types of diets. In addition, subjects were given an 8-week low-carbohydrate meal plan or reduction diet meal plan as per randomization.

### 2.3. Training Protocol

#### 2.3.1. Development of Strength Skills

The plan was designed to develop maximum strength in the tested exercise and the muscles involved. 3 differently focused trainings per week were performed:Focused on chest—bench press.Focused on the muscles of the lower limbs—leg press.Focused on the back muscles—lat pull-down.

One training unit lasted approximately 60 min. For each training unit, the full focus was on the technique of execution and time under tension. Each training unit was performed with the maximum possible effort to achieve the maximum results. The prescribed intensity in the form of load was individualized and based on the entry measurements. The technical design, time under tension and maximum effort must were similar for all subjects (maximum effort = maximum possible intensity in compliance with technical parameters and number of repetitions) under tension and maximum effort were similar for all subjects (maximum effort = maximum possible intensity in compliance with technical parameters and number of repetitions)

#### 2.3.2. Development of Endurance Skills

The plan consisted of a 30-min run at constant heart rate (at approximately 70% max TF or around 130–140 heart beats/minute).

#### 2.3.3. Supervision of Adherence to Training and Diet Protocols

Overall adherence to diet was checked once weekly by a nutritionist. Furthermore, adherence to CKD was evaluated through urinary ketone measurements performed twice daily and by measurement of blood β-hydroxybutyrate at the end of the study.

Training compliance was monitored through mandatory check-in procedures in a gym, and also by a sport tester for aerobic performance (TomTom Runner Cardio, TomTom, The Netherlands).

### 2.4. Post-Intervention Testing

Data collection procedures were the same as baseline testing procedures. To ensure reliability, power measures and performance testing were completed by the same researcher as at baseline for each subject. In addition, subjects conducted their testing at the same time and with the same personal researcher as pre-testing. Results from all tests were compared to the individual’s baseline values and provided to the subjects after data analysis.

### 2.5. Statistical Analysis

Statistical analysis was performed using Sigma Stat software (SPSS Inc., Chicago, IL, USA). Graphs were drawn using SigmaPlot 13.0 software (SPSS Inc., Chicago, IL, USA). The results are expressed as mean ± standard deviation (SD). Differences of body composition (body fat %, weight, BMI, lean body mass, and fat mass), biochemical, and strength or aerobic performance parameters between CKD and RD were evaluated using one-way ANOVA followed by Holm-Sidak test. Paired t-test was used for the assessment of intra group differences as appropriate. Statistical significance was assigned to *p* < 0.05.

## 3. Results

### 3.1. The Influence of Cyclical Ketogenic Reduction Diet vs. Nutritionally Balanced Reduction Diet on Anthropometric and Biochemical Parameters

Both CKD and RD decreased body weight (Figure 2), body fat mass and body mass index with comparable effects of both approaches (Table 1). Lean body mass and body water content was significantly reduced by CKD (Figure 3 and Figure 4 and Table 1) while it was not influenced by RD.

None of the diets significantly affected serum concentration of creatine kinase or lactate dehydrogenase (Table 1), liver tests, urea, creatinine, or circulating lipids (data not shown). β-hydroxy-butyrate significantly increased in CKD group while it was unaffected in subjects on reduction diet (Table 1).

### 3.2. The Influence of Cyclical Ketogenic Reduction Diet vs. Nutritionally Balanced Reduction Diet on Muscle Strength Parameters

The muscle strength parameters were assessed as maximum weight lifted during bench press, lat pull-down, and leg press. CKD did not affect any of these parameters (Table 2). On the contrary, in subjects on RD lat pull-down and leg press values significantly increased (Table 2).

### 3.3. The Influence of Cyclical Ketogenic Reduction Diet vs. Nutritionally Balanced Reduction Diet on Spiroergometric Parameters

Spiroergometric parameters are shown in Table 3. Respiratory exchange ratio decreased in subjects on CKD while it did not change in subjects on RD. None of other spiroergometric parameters were significantly affected in CKD group.

In contrast, in RD group peak workload, peak oxygen uptake/kg, peak workload/kg, and physical working capacity at a heart rate of 170/min increased after 8 weeks of intervention.

## 4. Discussion

The most important finding of this study is that eight weeks of regular aerobic exercise combined with exercise training complemented by two different dietary approaches—cyclical ketogenic reduction diet or nutritionally balanced reduction diet—significantly decreased body weight and body fat in healthy young men to a similar degree while having differential influence on body composition, strength parameters, and aerobic performance.

Despite comparable influence of both diets on body weight, we detected distinctions in their effects on body composition. In CKD group, the drop of body weight was due to a combination of decreased body fat, body water, and a slight, but significant, decline in lean body mass. On the contrary, in RD patients neither body water nor lean body mass were significantly affected and the weight reduction was predominantly due to body fat loss. The influence of ketogenic diet combined with different forms of exercise on body composition has been studied both in athletes and in patients with obesity and other comorbidities on numerous occasions. In some of the trials, isocaloric [24] or hypocaloric ketogenic diet [25] did not significantly change lean body mass while reducing body fat. On the contrary and in agreement with our current data, Perissious and colleagues found a reduction in lean body mass in patients with obesity undergoing exercise program while being on low carbohydrate diet [26]. Differential effect of ketogenic vs. nutritionally balanced diet under hyperenergetic conditions has also been described in a study in healthy men undergoing an eight-week resistance training program. Under these conditions, lean body mass increased only in control diet while it was unaffected in the ketogenic diet group [27]. Finally, ad libitum low carbohydrate ketogenic diet reduced body mass and lean body mass without compromising performance in powerlifting and Olympic weightlifting athletes [28]. In our study, a slight decrease in lean body mass did not impair strength parameters as compared to baseline values. Nevertheless, we have noted that in RD patients both lat pull-down and leg-press significantly increased after eight weeks of intervention as compared to no change in subjects on CKD.

While neutral effect of CKD on strength parameters in our study could have been expected based on the previously published data [29,30], we hypothesized that ketogenic diet could be more efficacious in improving endurance parameters as compared to nutritionally-balanced reduction diet as suggested by some previous trials [31]. The increasing popularity of ketogenic diets in endurance athletes is based on the hypothesis that predominant fat utilization over the use of carbohydrates may improve energy availability during endurance exercise along with accelerated recovery [10]. Bailey and Hennesy recently reviewed available data on the influence of ketogenic diet on endurance in athletes. They included seven studies into their analysis and concluded that limited and heterogenous findings prohibit definitive conclusions [16]. In our study, we found decreased respiratory exchange ratio in CKD groups after eight weeks of intervention as compared with no effect of RD suggesting a shift towards lipid oxidation, which is in agreement with the mode of action of ketogenic diet and previously published data [32]. However, none of the endurance parameters as measured by spiroergometry have been affected in CKD group. On the contrary, in RD group peak oxygen uptake and peak workload significantly increased after eight weeks of intervention. Our data suggesting lack of improvement of endurance performance by ketogenic diet go in similar direction with results published by Burke and colleagues in 2017 [33] and reproduced by the same group in 2020 [34] where they found decreased endurance parameters in elite race walkers after ketogenic diet. By contrast, in one of the early studies, low carbohydrate diet improved endurance times during moderate exercise in moderately obese patients along with significant reductions in body weight and body fat mass [35]. Nevertheless, despite more pronounced fat loss the improvement on endurance performance with low carbohydrate diet was comparable to that of high carbohydrate diet group.

When interpreted the results of our study within the context of currently published data it is important to consider its strengths and limitations. The randomized design and the good compliance of the subjects to dietary and treatment regimens can be consider strong points of our trial. On the other hand, the limitations include relatively short duration, low number of subjects and inclusion of only male participants.

Taken together our data are in general agreement with most of the previously published studies [36] showing little or no benefit of ketogenic diet on endurance capacity. However, it should be noted that contribution of fatty acids to metabolic response may differ with respect to duration and intensity of exercise [37,38], exact type of training and numerous other characteristics. The utilization of fatty acids increases with prolonged bouts of exercise of moderate intensity suggesting that ketogenic diet might be useful especially with longer duration of aerobic exercise.

## 5. Conclusions

In summary, our data show that in healthy young males undergoing resistance and aerobic training comparable weight reduction can be achieved with ketogenic and nutritionally balanced reduction diet. In RD group, improved muscle strength and endurance performance was noted relative to neutral effect of CKD on these parameters. Furthermore, CKD also slightly reduced lean body mass. Our study thus demonstrates that the cyclical ketogenic reduction diet effectively reduces body weight but is not an effective strategy to increase anaerobic or strength performance in healthy young men. All in all, further randomized studies of longer duration are still needed to explore whether the response to different diets is affected by long-term adaptation responses and whether it differs in males and females or subjects with various types and levels of fitness.

## Figures and Tables

**Figure 1 nutrients-12-02832-f001:**
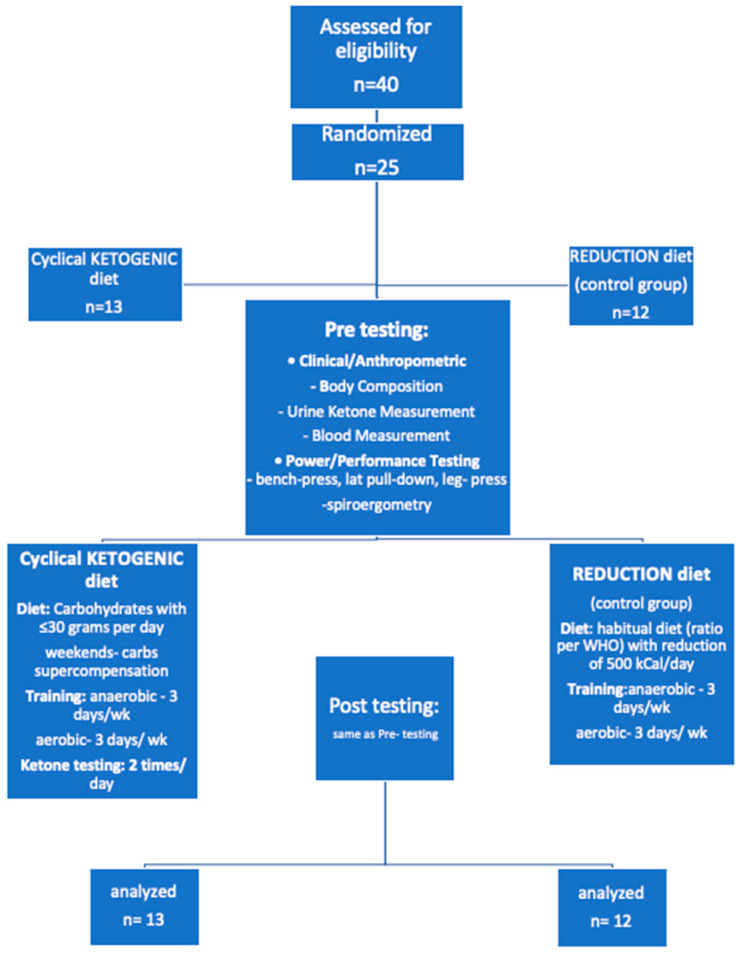
CONSORT diagram of subjects participating in an 8-week program while consuming a cyclical ketogenic reduction diet (CKD) or nutritionally balanced reduction diet (RD).

**Figure 2 nutrients-12-02832-f002:**
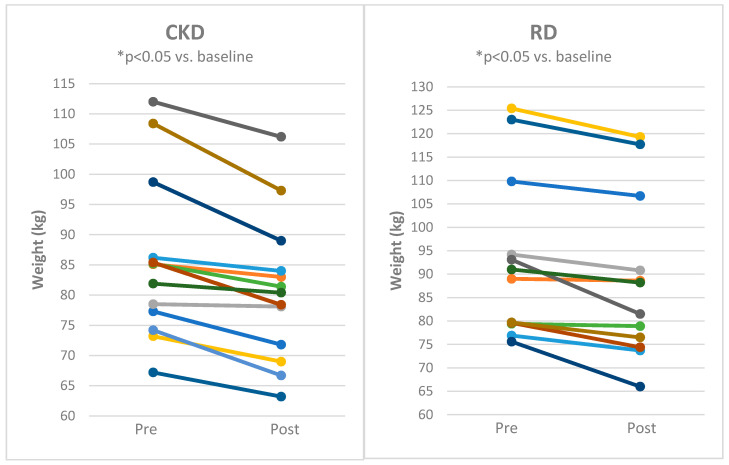
Individual responses of body weight for subjects before and after 8 weeks of cyclical ketogenic reduction diet (CKD) and nutritionally balanced reduction diet (RD). Statistical significance is from paired *t*-test * *p* < 0.05 vs. baseline.

**Figure 3 nutrients-12-02832-f003:**
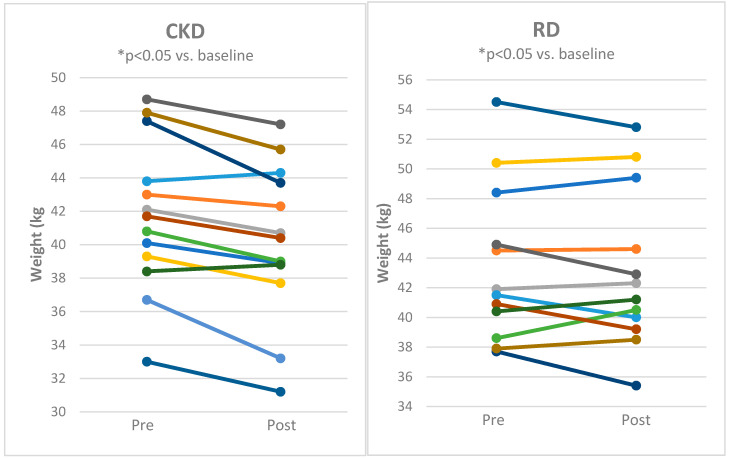
Individual responses of lean body mass for subjects before and after 8 weeks of cyclical ketogenic reduction diet (CKD) and nutritionally balanced reduction diet (RD). Statistical significance is from paired t-test * *p* < 0.05 vs. baseline.

**Figure 4 nutrients-12-02832-f004:**
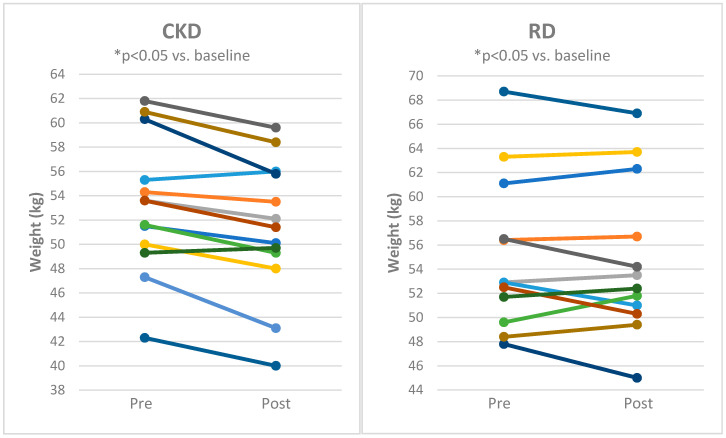
Individual responses of body water weight for subjects before and after 8 weeks of cyclical ketogenic reduction diet (CKD) and nutritionally balanced reduction diet (RD). Statistical significance is from paired t-test * *p* < 0.05 vs. baseline.

**Table 1 nutrients-12-02832-t001:** Anthropometric and biochemical parameters of subjects on cyclical ketogenic reduction diet or nutritionally balanced reduction diet at baseline and after 8 weeks of diet.

	Cyclical Ketogenic Diet (CKD)		Reduction Diet (RD)		ANOVA
	V1-before	V2-after	V1-before	V2-after	
Number (*n*)	13	13	12	12	
Age (year)	23 ± 5	NA	24 ± 4	NA	NS
Height (cm)	181 ± 6	NA	186 ± 10	NA	NS
BMI (kg/m^2^)	26.1 ± 3.7	24.6 ± 3.3 *	26.9 ± 4.3	25.5 ± 4.2 *	NS
WEIGHT (kg)	85.6 ± 13.4	81.0 ± 12.0 *	93.0 ± 17.5	88.5 ± 17.4 *	NS
MUSCLES (kg)	41.8 ± 4.5	40.0 ± 4.6 *	43.5 ± 5.3	43.1 ± 5.3	NS
FAT (kg)	12.9 ± 6.9	11.0 ± 5.8 *	17.6 ± 9.8	13.6 ± 9.0 *	NS
% FAT	14.5 ± 5.5	13.0 ± 5.1 *	17.9 ± 6.9	14.2 ± 6.9 *	NS
WATER (kg)	53.2 ± 5.6	51.0 ± 5.6 *	55.1 ± 6.4	54.8 ± 6.5	NS
CK (ukat/L)	4.40 ± 2.81	2.81 ± 1.21	3.80 ± 2.03	3.03 ± 2.03	NS
LDH (ukat/L)	2.68 ± 0.60	2.47 ± 0.42	2.74 ± 0.44	2.55 ± 0.33	NS
β-OH-butyrate (mmol/L)	0.2 ± 0.07	0.38 ± 0.25 *	0.24 ± 0.12	0.12 ± 0.04	NS

Data are mean ± SD. Statistical significance is from One-way ANOVA and paired *t*-test (V1—baseline testing vs. V2—testing after 8 weeks of diet). * *p* < 0.05 vs. V1. BMI: Body mass index; CK: Creatine kinase; LDH: Lactate dehydrogenase; β-OH-butyrate—β-hydroxy-butyrate. NS: Not significant. NA : not avalible.

**Table 2 nutrients-12-02832-t002:** The effect of cyclical ketogenic reduction diet and nutritionally balanced reduction diet on strength parameters.

	Cyclical Ketogenic Diet (CKD)		Reduction Diet (RD)		ANOVA
	V1-before	V2-after	V1-before	V2-after	
Bench press (BP)	90.0 ± 24.2	90.0 ± 23.7	84.2 ± 21.8	87.7 ± 20.1	NS
Lat pull-down (LPD)	74.2 ± 15.7	76.0 ± 15.0	70.4 ± 14.8	75.2 ± 17.1 *	NS
Leg press (LP)	138.0 ± 21.1	142.0 ± 16.3	127.8 ± 22.0	140 ± 22.8 *	NS

Data are mean ± SD. Statistical significance is from One-way ANOVA and paired *t*-test (V1—baseline testing vs. V2—testing after 8 weeks of diet). * *p* < 0.05 vs. V1. NS: Not significant. NA : not avalible.

**Table 3 nutrients-12-02832-t003:** The effect of cyclical ketogenic reduction diet and nutritionally balanced reduction diet on aerobic performance parameters.

	Cyclical Ketogenic Diet (CKD)		Reduction Diet (RD)		ANOVA
	V1-before	V2-after	V1-before	V2-after	
TFmax	180.9 ± 10.2	178.0 ± 11.3	178.9 ± 11.8	179.0 ± 10.2	NS
Rmax	1.27 ± 0.08	1.2 ± 0.12 *	1.21 ± 0.04	1.16 ± 0.10	0.04
Wmax	297.0 ± 48.5	298.0 ± 54.3	282.1 ± 34.3	296.0 ± 35.9 *	NS
VEmax	121.0 ± 28.5	136.0 ± 30.0	113.2 ± 20.3	124.0 ± 21.3	NS
VO_2_max/kg	40.2 ± 4.1	43.0 ± 5.4	35.2 ± 6.0	38.2 ± 6.3 *	0.007
VO_2_max/TF	19.0 ± 3.3	20.0 ± 3.4	18.0 ± 1.9	18.9 ± 1.6	NS
Wmax/kg	3.53 ± 0.42	3.6 ± 0.39	3.13 ± 0.52	3.36 ± 0.59 *	NS
W170max/kg	3.27 ± 0.65	3.4 ± 0.37	2.8 ± 0.74	3.06 ± 0.83 *	NS

Data are mean ± SD. Statistical significance is from One-way ANOVA and paired *t*-test (V1—baseline testing vs. V2—testing after 8 weeks of diet). * *p* < 0.05 vs. V1 TFmax—maximal heart rate; Rmax—respiratory exchange ratio; Wmax—peak workload; VEmax—maximal pulmonary ventilation; VO_2_ max/kg—peak oxygen uptake; VO_2_max/TF—peak pulse oxygen; Wmax.kg peak workload/kg; W170 max/kg—physical working capacity (at a heart rate of 170/min). NS: Not significant. NA : not avalible.

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
