# Peer review of "The Influence of Cyclical Ketogenic Reduction Diet vs. Nutritionally Balanced Reduction Diet on Body Composition, Strength, and Endurance Performance in Healthy Young Males: A Randomized Controlled Trial"

_nutrients, 2020, doi:10.3390/nu12092832_

Round 1

Reviewer 1 Report

This research is well done and the methodology and results are well structured and presented.

Regarding the paper itself, in my opinion, some minor modifications are required in order to facilitate the reader’s understanding of the research process and information presented.

I believe that the authors chose the modality of presenting the results by means and SD according to the data distribution (normal). If the data distribution is normal, I do not understand the remark of Wilcoxon Signed-Rank test utilization, which is a nonparametric test.

Please resolve the inadvertency that appears in the Biochemical and anthropometric examination section, namely that the determination of liver tests, urea, creatinine, or circulating lipids are not mentioned like in the results section (only that “the liver tests, urea, creatinine or circulating lipids – data not shown”).

I'd also suggest that the discussion section needs to be more exhaustive.

In addition to this, I believe that highlighting some limits and strong points would be welcomed.

Finally, for further improving the Introduction section, I invite the authors to read and consider a relevant paper such as:

Kelly, T.; Unwin, D.; Finucane, F. Low-Carbohydrate Diets in the Management of Obesity and Type 2 Diabetes: A Review from Clinicians Using the Approach in Practice. Int. J. Environ. Res. Public Health 202017, 2557.

Reviewer 2 Report

Dear Authors thanks for your valuable contribution.

The title and the abstract are appropriate and clearly state the key features of this work. The background is well explained, providing key information on the scientific topic, which is correctly supported by references. The aims of the research are clearly stated. The participant's selection process is clearly described as well as the ethical requirements, which are met. Discussion and conclusions are clearly presented. The material and methods section needs to be improved.

Here attached some minor points:

  • Line 88: (Ethics approval) please add the protocol number
  • Line 93: the authors stated that a 500 kcal/day reduction was adopted without explaining why. Please provide more details
  • Line 107: BMI was calculated by the scale? Using the height measurement too? Please clarify it adding height measurement details too.
  • Line 111: some biochemical parameters have been chosen and analysed; however, the authors did not justify this choice (what each parameter does stand for?) and did not properly describe the sampling collection procedure (which media? When? How was collected? How long was the sample stored for? Temperature? Please provide laboratory methods description and units of measure too.
  • Lines 125-131: how was the strength testing recorded? (The authors stated it in another section, I would suggest putting it here too).
  • Figure 2: please revise the titles (I would suggest removing “weight” from the title and put it in the proper axes (y); the same for lean body mass and body water graphs. Please add the p-value and the statistical test too.
  • Line 251: “subjects” instead of “patients”
  • Line 275: I would suggest adding “combined with exercise training” after “diet-“or were fits better in the sentence.
  • Please add the strengths and limitations of this study in the discussion section and future research trends suggestions in the conclusions.
  • Please add the trial registration number.

Best regards
